# Identification of Candidate Genes for Salt Tolerance at the Seedling Stage Using Integrated Genome-Wide Association Study and Transcriptome Analysis in Rice

**DOI:** 10.3390/plants12061401

**Published:** 2023-03-21

**Authors:** Tae-Heon Kim, Suk-Man Kim

**Affiliations:** 1Institute of Agricultural Science and Technology, Kyungpook National University, Daegu 41566, Republic of Korea; kimth6148@knu.ac.kr; 2Department of Ecological & Environmental System, Kyungpook National University, Sangju 37224, Republic of Korea

**Keywords:** genome-wide association study, RNA-Seq, rice, salt stress, seedling stage

## Abstract

Salt stress is a major constraint in rice production worldwide. Salt stress is estimated to cause annual losses of 30–50% in rice production. Discovering and deploying salt-resistance genes are the most effective ways to control salt stress. We performed a genome-wide association study (GWAS) to detect QTLs related to salt tolerance at the seedling stage using the *japonica*-multiparent advanced generation intercross (MAGIC) population. Four QTLs (*qDTS1-1*, *qDTS1-2*, *qDTS2*, and *qDTS9*) associated with salt tolerance were identified on chromosomes 1, 2, and 9. Among these QTLs, a novel QTL, *qDTS1-2*, was located between flanking SNPs (1354576 and id1028360) on chromosome 1, with the largest −log10(P) value of 5.81 and a total phenotypic variance of 15.2%. RNA-seq analysis revealed that among the seven differentially expressed genes (DEGs) commonly identified in both P6 and JM298 showing salt tolerance, two upregulated genes, *Os01g0963600* (ASR transcription factor) and *Os01g0975300* (*OsMYB48*), related to salt and drought tolerance, were also involved in the target region of *qDTS1-2*. The results of this study can provide insights into further understanding of salt tolerance mechanisms and developing DNA markers for marker-assisted selection (MAS) breeding to improve the salt tolerance of cultivars in rice breeding programs.

## 1. Introduction

Salt stress is a major constraint in rice production worldwide [1,2]. Over 6% of the land area in the world is facing salinity problems, and particularly in Asia, 21.5 million hectares of arable land are damaged by salt [3,4]. In recent years, a considerable proportion of arable land has become saline due to excessive irrigation and sea level rise in coastal regions related to climate change [4]. Rice yield has been reported to be significantly reduced by an estimation of 30–50% in salt-affected areas [5,6].

Salt stress (or salinity) refers to problems occurring in soil due to the presence of excessive salt; these soils are categorized as sodic and saline soils [7]. Sodic soils occur mainly in arid and semiarid regions and have a high pH (>8.5) and a high exchangeable sodium percentage (>15) in colloidal surfaces. Saline soils are usually found in arid and coastal regions and contain neutral salts, such as chlorides (Cl^−^) and sulfates (SO_4_^2−^) of sodium (Na^+^), calcium (Ca^2+^), magnesium (Mg^2+^), and potassium (K^+^), with electrical conductivity of >4 dSm^−1^.

Salt stress causes damage to the plants at the seedling stage, leading to whitened tips of leaves, chlorotic patches on leaves, plant stunting, reduced tillering, and death [8]. Two main mechanisms of salt stress reduce plant growth and productivity: (1) Osmotic stress reduces water uptake from roots and causes dehydration, similar to drought stress. (2) Accumulation of salts leads to ion toxicity, which causes a decrease in photosynthetic abilities and damages plant cells.

The development of salt-tolerant cultivars employing tolerance genes is the most effective method for controlling salt stress [9]. However, a narrow pool of genetic resources has been identified as a salt tolerance donor in rice [10,11], and limited research has been done on it in *japonica* rice. Salt tolerance is a complex trait controlled by multiple genes [12], and its detection via screening methods is difficult. Therefore, improving salt tolerance in rice has been challenging until now. Nevertheless, the requirements for developing rice cultivars related to stability in rice production have increased, especially in saline, reclaimed land, and coastal areas [12,13,14].

Recently, according to the availability of rice genome sequences, a great number of molecular markers has developed, precisely located on rice chromosomes. QTL mapping has provided insights into the inheritance mechanisms for salt tolerance using molecular markers in rice [15,16]. Numerous studies have identified QTLs associated with salt tolerance. *Saltol*, a major QTL associated with K^+^/Na^+^ homeostasis in the shoot, was mapped on chromosome 1 using recombinant inbred lines (RILs) derived from a cross between IR29 and Pokkali [17,18]. *SKC1*, which controls the K^+^ concentration in the shoot derived from Nona Bokra landrace, was mapped in the same region of *Saltol* [18] and later cloned as *OsHKT1;5* encoding a Na^+^-selective transporter [19,20]. The *Saltol* locus was transferred into several popular salt-tolerant varieties, such as Rassi, Pusa Basmati 1, and Pusa Basmati 101, via marker-assisted breeding [13,21,22,23]. In addition, QTLs were reported in various traits associated with salt tolerance at the seedling stage, such as the degree of damage symptoms and morpho-physiological parameters. Javed et al. [24] identified 10 QTLs associated with the salt injury score, seedling height, shoot weight, and shoot dry weight on chromosomes 1, 3, 6, 7, 8, and 11, with the phenotypic variation explained (PVE) for 4.9–9.3%. They revealed that the salt injury score was negatively correlated with the sodium and potassium contents in the roots, but positively correlated with the sodium and potassium content ratio in the shoots. In BC_3_F_4_ introgression lines of Pokkali without the *Saltol* locus, 6 QTLs for the salt injury score were mapped on chromosomes 1, 3, 4, 10, and 11, with a PVE of 5.3–18.4% [25]. Using 187 RILs derived from the At354 × Bg352 cross, 4 QTLs, i.e., *qSS1*, *qSS4*, *qSI1*, and *qSI4*, were identified for the salinity survival index and shoot length on chromosomes 1 and 4, with a PVE of 10.0–15.0% [26]. De Leon et al. [27] identified 38 QTLs related to the injury score, chlorophyll, shoot length, root length, dry weight, and shoot-root ratio on all chromosomes except chromosome 10, with a PVE of 2.1–51.6%, using 187 RILs developed from a cross between Bengal and Pokkali. They suggested candidate genes associated with traits using a high-density linkage map. For the salt tolerance score and the survival rate under salt stress, Wang et al. [28] identified 16 QTLs derived from *O. rufipogon* on chromosomes 1, 5, 7, and 9–12, with a PVE of 2–8% using the RIL population. They reported that the candidate genes *LOC_Os05g31620* and *LOC_Os10g34730* were related to salt tolerance, as determined using qRT-PCR and RNA-seq analysis.

Genome-wide association studies (GWASs) are a powerful approach for identifying valuable natural variations in trait-associated loci based on high-density genomic information and allelic variations in candidate genes underlying quantitative and complex traits, including salt tolerance [29,30]. Development of next-generation sequencing (NGS) techniques and completion of high-quality reference genome sequences led to the initiation of GWAS [31]. NGS generates large amounts of genomic information that can detect genetic variants with high throughput and low cost [32]. Furthermore, transcriptome analysis combined with GWAS is helpful for discovering genes and clarifying the roles of biological pathways and mechanisms related to salt tolerance [14]. Zhang et al. [33] identified 7 QTLs on chromosomes 1, 2, 5, and 9 associated with salt tolerance using a multiparent advanced generation intercross (MAGIC) population, which explained 7.42–9.38% of the total phenotypic variation. Liu et al. [30] identified five known genes (*OsMYB6*, *OsGAMYB*, *OsHKT1;4*, *OsCTR3*, and *OsSUT1*) and two novel genes (*LOC_Os02g49700* and *LOC_Os03g28300*) significantly associated with grain yield and its related traits under saline stress conditions.

Neang et al. [34] conducted GWAS using 296 accessions of rice and 36,901 SNPs to identify the significant SNPs associated with the Na^+^ sheath:blade ratio. They found substantial SNPs only in the *indica* subpopulation on chromosome 5. Moreover, they found 13 candidate genes, including 5 upregulated and 8 downregulated genes, in the internal leaf sheath tissues in the presence of salt stress. Using genome-wide association mapping, Li et al. [29] identified eight QTLs significantly associated with alkalinity tolerance at the seedling stage. They confirmed that *OsIRO3*, encoding a bHLH-type transcription factor, was the candidate gene and a negative regulator of the Fe-deficiency response in rice. Le et al. [35] screened 9 salinity tolerance-related traits, including the salt injury score, chlorophyll, water content, and K^+^ and Na^+^ contents, using 179 Vietnamese rice landraces to identify novel QTLs. A total of 26 QTLs were identified using GWAS, of which 10 were associated with hormone transduction pathways or transcriptional modulation of gene expression in response to stress, suggesting that these QTLs act in complementary ways to control salinity tolerance.

Herein, we performed a GWAS analysis to identify QTLs related to salt tolerance at the seedling stage using the *japonica*-MAGIC population. RNA-Seq analysis was conducted to confirm the candidate genes associated with salt tolerance at the seedling stage located in the target regions of QTLs. The candidate genes identified in this study might be useful for the future development of salt-tolerant *japonica* breeding lines via marker-assisted selection (MAS).

## 2. Results

### 2.1. Salt Tolerance at the Seedling Stage in the JMAGIC Population

The salt tolerance of the parents and JMAGIC was evaluated at the seedling stage to obtain the phenotype data. P6, one of the parents of JMAGIC, was moderately tolerant to salt stress, with a DTS of 5.0 ± 0.7 (Table 1 and Appendix A, Figure 1), but the other 7 parents were susceptible to salt stress, with a DTS of 5.8–9.0. The salt tolerance of P6 was similar to that of Pokkali (4.8 ± 1.1) (Appendix A) used as a tolerance check. A total of 221 lines with stable reactions in the replicate evaluation tests under salt stress from 381 JMAGIC lines were selected as a mapping population for further study in the evaluation of salt tolerance. The mapping population showed a DTS of 4.8–9.0, with a mean of 8.1 under salt stress conditions. The DTS was not normally distributed and biased to susceptibility, with a skewness of −1.6 (Figure 2).

### 2.2. GWAS of Salt Tolerance

The SNPs were filtered to obtain the genotype data with parameters of MAF > 0.05 and a missing rate <30%. Eventually, 6356 SNPs were selected. GWAS analysis was performed using the mixed linear model (MLM) of GAPIT in the mapping population using the phenotype data and genotype data. The significant SNPs associated with salt tolerance were illustrated in whole chromosomes after the analysis (Figure 3). Six significant SNPs associated with DTS were detected on chromosomes 1, 2, and 9. Among the six SNPs, three were located on chromosome 1, two on chromosome 9, and one on chromosome 2 (Table 2 and Figure 3). The significant SNPs located on chromosomes 1, 2, and 9 were designated as *qDTS1-1*, *qDTS1-2*, *qDTS2*, and *qDTS9*, respectively. A total of 4 QTLs associated with DTS showed a −log10(P) value of 3.34–5.81, with a PVE of 7.65–15.20% (Table 2). *qDTS1-1* was located at 8.38 Mb on chromosome 1, showing a −log10(P) value of 3.34, with the smallest PVE value of 7.65%. In contrast, the peak SNP of *qDTS1-2*, located at 42.72 Mb on chromosome 1, explained the largest PVE value of 15.20% for DTS, with a −log10(P) value of 5.81. *qDTS2*, located at 22.25 Mb on chromosome 2, showed a −log10(P) value of 3.42, with a PVE value of 7.87%. The 2 peak SNPs of *qDTS9* located on chromosome 9 showed identical −log10(P) and PVE values of 3.94% and 9.41%, respectively. The peak SNP of *qDTS1-1* was located within *Os01g0253300*, encoding the importin alpha-1a subunit. The peak SNP of *qDTS1-2* was located within *Os01g0968600*, encoding the leucine-rich repeat family protein. The peak SNP of *qDTS9* was located within *Os09g0299200*, encoding the MYB-type transcription factor.

### 2.3. Candidate Genes Associated with Salt Tolerance

All annotated genes included in the target region were extracted using RAP-DB (https://rapdb.dna.affrc.go.jp, accessed on 2 February 2023) to identify the candidate genes associated with salt tolerance at the seedling stage. In the 4 target regions of the QTLs, 329 annotated genes were selected (Appendix A). Among the genes, the results indicated that each target region on chromosomes 1, 2, and 9 contained 208, 80, and 41 annotated genes, respectively.

### 2.4. Expression Analysis Via RNA-Seq

Two lines (P6 and JM198) were selected to identify the differentially expressed genes (DEGs) associated with salt tolerance, showing the most reliable responses to salt stress in this test. One line of the JMAGIC population, JM198, has a DTS of 4.8 (data not shown). RNA-Seq analysis was performed using the lines to compare the expression levels of the candidate genes before and after salt stress treatments (Appendix A). A total of 1532 DEGs were identified, with a mean value of 128 across 12 chromosomes in P6 (Appendix A), and in the case of JM198, a total of 1,290 DEGs were identified across 12 chromosomes, with a mean value of 108. The DEGs commonly identified in both lines were 665 out of 2,822 DEGs. Among the 665 DEGs, 22 were involved in the 4 loci associated with salt stress (Figure 4 and Appendix A). In particular, the seven overlapping DEGs validated the target regions on chromosomes 1, 2, and 9 (Table 3). The expression of two DEGs among the seven was upregulated and that of five DEGs was downregulated under salt stress. The results showed that the downregulated gene *Os01g0253900*, which encoded triacylglycerol lipase, was located in the *qDTS1-1* region, and two upregulated genes (*Os01g0963600* and *Os01g0975300*) along with three downregulated genes (*Os01g0962700*, *Os01g0971800*, and *Os01g0974200*) were identified in the *qDTS1-2* region. The five DEGs encoded the abscisic acid, stress, and ripening (ASR) transcription factor (*Os01g0963600*); MYB-related transcription factor (*Os01g0975300*); peroxidase 12 precursors (*Os01g0962700*); a transcription factor with a GARP DNA-binding domain (*Os01g0971800*); and metallothionein (*Os01g0974200*). The downregulated *Os09g0294000* was located in the *qDTS9* region and encoded bifunctional aspartokinase/homoserine dehydrogenase 2. In contrast, there were no DEGs in the *qDTS2* region.

## 3. Discussion

Salt stress is a major constraint in rice production and restricts the use of agricultural land [9,36]. Moreover, salt stress is progressively increasing due to human activity and climate change [37]. For stability in rice yield products, the requirements have increased for the development of salt-tolerant rice cultivars, especially in saline reclaimed land and coastal areas, despite challenges due to the complexity combined with the difficulty in phenotyping [12,13,14]. In this study, we performed GWAS analysis and RNA-Seq analysis to detect QTLs related to salt tolerance at the seedling stage using the MAGIC population.

The salt tolerance of eight parents and the JMAGIC population was evaluated with a tolerance check and susceptible check (Figure 1 and Figure 2). Only P6 showed stable moderate tolerance among the parents (5.0 ± 0.7) (Figure 1 and Appendix A). Others did not show tolerant reactions to salt stress during the replication tests. Bandillo et al. [38] stated that P1 is the Basmati rice group, with sodicity tolerance originating from India. In this study, P1 showed a susceptible reaction with 9.0 ± 0.7 DTS under salt stress conditions. It is assumed that the salt tolerance mechanism involved in P1 differed from that in P6 with salinity tolerance. Therefore, we could confirm P6 as a source has alleles/genes related to salt tolerance conferring tolerance to salt stress in the JMAGIC population. The 221 lines that showed consistent tolerant reactions in the replicate evaluation tests under salt stress were selected for a mapping population from the 381 JMAGIC lines. The other 160 lines that showed high variations in DTS were excluded to identify the reliable QTLs related to salt tolerance using GWAS analysis. The frequency distribution of DTS in the population showed a range of 4.8–9.0, with a mean of 8.1, under salt stress conditions (Appendix A). This tendency was similar to the previous study, which was not normally distributed for salt tolerance at the seedling stage (Figure 2) [39].

GWAS analysis was performed using the phenotype data to identify the QTLs related to salt tolerance at the seedling stage in the population. From the GWAS analysis, six significant SNPs associated with DTS were detected on chromosomes 1, 2, and 9, and the SNPs for salt tolerance at seedlings were designated as *qDTS1-1*, *qDTS1-2*, *qDTS2*, and *qDTS9* (Table 2 and Figure 3). In the significant SNPs, SNP_IDs 257670 and 1354576 were considered a single QTL, *qDTS1-2*, because they were located at a region of less than 300 kb on chromosome 1, and SNP_IDs 257670 and 1354576 located on chromosome 9 were also considered a single QTL, *qDTS9*, for the same reason. *qDTS1-2*, explaining the largest −log10(*p*) value of 5.81 for DTS, with a PVE value of 15.20%, was located in *Os01g0968600*, encoding the leucine-rich repeat family protein. The function of *Os01g0968600* was unknown. Moreover, *qDTS1-2* was closely located to the salt tolerance genes *OsP5CR*, *OsMSR2*, and *OsMYB48* (Figure 5). In the case of *OsP5CR*, it is a crucial gene for proline biosynthesis, related to the osmotic regulator in rice [40,41]. *OsMSR2*, encoding a novel calmodulin-like protein, enhanced the tolerance of salt and drought through ABA-mediated pathways, increased ABA sensitivity in *Arabidopsis*, and was also involved in the salt stress response in rice [14,42]. Xiong et al. [43] reported that *OsMYB48* was a novel MYB-related transcription factor positively affecting drought and salinity tolerance by regulating stress-induced ABA synthesis. Additionally, the peak SNP of *qDTS9* was located within *Os09g0299200*, encoding the MYB-type transcription factor (*OsMYBc*) regulating the expression level of *OsHKT1;1*, which has a role in controlling the Na^+^ concentration and preventing sodium toxicity in leaf blades under salt stress conditions [44]. *qDTS9* was closely located to the salt tolerance genes *OsNHX5*, *OsMYBc*, and *OsbZIP71*. *OsNHX5* plays an essential role in the compartmentation of the Na^+^ and K^+^ accumulating in the cytoplasm into the vacuole, according to Fukuda et al. [45]. *OsMYBc* and *OsbZIP71* are transcription factors related to salt tolerance, regulating the expression of *OsHKT1;1* (Na^+^ transport) and *OsNHX1* (Na^+^/H^+^ antiporters), respectively [46,47].

RNA-Seq was conducted using P6 and JM198 to identify the DEGs associated with salt tolerance in the target regions. The results showed that seven genes were finally selected as overlapping DEGs. Out of the seven genes, two were upregulated and five were downregulated. The upregulated two genes were *Os01g0963600* (ASR transcription factor) and *Os01g0975300* (*OsMYB48*) and were known to be related to salt and drought tolerance in rice [43,48,49]. In general, transcription factors are essential in linking salt sensory and regulating the expression levels of various genes that may eventually influence the level of salt tolerance in plants [50]. Considering these facts, we could suggest the most promising candidate genes, *Os01g0963600* (ASR transcription factor) and *Os01g0975300* (*OsMYB48*), are associated with salt tolerance at the seedling stage in this study.

Complementation tests, selection of helpful haplotypes, and development of PCR-based DNA markers for MAS will be conducted in further studies. This result will be beneficial for determining the genetic mechanisms contributing to salt tolerance in rice and developing rice cultivars with improved salt tolerance in the breeding program.

## 4. Materials and Methods

### 4.1. Plant Materials

The JMAGIC population was used for association mapping studies related to salt tolerance at the seedling stage. The JMAGIC population, comprising a total of 381 lines, was developed by intercrossing eight parents: P1 (CSR 30), P2 (Cypress), P3 (IAC 165), P4 (Jinbu), P5 (WAB 56-125), P6 (IR73571-3B-11-3-K2), P7 (Inia Tacuari), and P8 (Colombia XXI). Pokkali, the most widely used genetic resource for improving salt tolerance in rice, was used as a tolerance check, and IR29 was used as a susceptible check for evaluating salt tolerance.

### 4.2. Evaluation of Salt Tolerance

For one replication of evaluation tests for salt tolerance, three pre-germinated seeds of each line and parents were sowed in three separate cells of a seeding box, respective-ly. All seedlings, which showed weak symptoms and retarded growth, were removed at eight DAS (days after sowing) before salt stress treatment. The salt stress treatment experiment was performed in a greenhouse at 20–28 °C. The healthy seedlings of each line and parents were subjected to 1/1000 Hyponex (KisanBio, Seoul, Republic of Korea) solution that contained 0.35% NaCl for 4 days in the first salt stress treatment. Subsequently, that was moved to a 1/1000 Hyponex solution that contained 0.70% NaCl for 14 days, the second salt stress treatment. The solution surface was covered with plastic plates to prevent excessive evaporation of the solution, and the solution was replaced regularly once every three days. All the evaluation tests for salt tolerance were conducted with seven replications. After 14 days of salt stress, the degree of salt tolerance at the seedling stage (DTS) for each line and parent seedlings was recorded as 1–9, according to the standard evaluation system (Table 1) [51].

### 4.3. DNA Extraction and High-Throughput SNP Genotyping

Genomic DNA was extracted from the fresh leaves of 21 DAS seedlings of 381 JMAGIC lines and parents using a Biosprint 96 DNA Plant Kit (Qiagen, Hilden, Germany), following the manufacturer’s protocol. The DNA concentration of each sample was measured using Nanodrop (Thermo Fisher Scientific, Waltham, MA, USA) and was adjusted to 50 ng/µL for high-throughput genotyping. SNP genotyping assays of the JMAGIC population and parents were performed using a 7K Infinium SNP genotyping platform at the Genotype Service Laboratory in the International Rice Research Institute, Philippines (IRRI). The 7K SNP BeadChip, an improved version of the Cornell_6K_Array_Infinium_Rice (C6AIR) [52], included 7098 SNPs consisting of 4007 SNPs from the 6K array [52], 2056 SNPs from the high-density rice array (HDRA) [53], 910 SNPs from the 384-SNP GoldenGate sets (OPA2.1, 3.1, 4.0, 5.0, 6.2, and 7.0) [54], 189 SNPs from the 44K array [55], and 21 gene-based SNPs. The Infinium II BeadChips, hybridized with amplified DNA, were stained using fluorescent dye. Subsequently, the fluorescence intensity of the BeadChips was measured by the BeadArray Reader. The raw fluorescence intensity values were converted to SNP data using GenomeStudio software(version 2011.1; Illumina, San Diego, CA, USA).

### 4.4. GWAS

Association analysis was conducted using the MLM implemented in the GAPIT. The MLM model incorporates the kinship (K) matrix and the population structure matrix (Q) to account for the relatedness among the genotypes and reduce the false positives that arise from family relatedness [56]. The genotype data obtained from the 7K SNP BeadChip were filtered with the following criteria: minor allele frequency (MAF) > 0.05 and missing rate <30%. The parameter of the PCA, total = 3, was applied to infer the population structure. The adjacent significant SNPs were considered a single association locus or QTL following the previously reported genome-wide linkage disequilibrium decay rates [57,58,59]. Additionally, all significant SNPs located in the chromosomal regions were considered significant QTLs. The region 300 kb upstream and downstream of the significant SNPs was considered the chromosomal positions for selecting the candidate genes. The candidate genes were searched within these regions using RAP-DB (https://rapdb.dna.affrc.go.jp, accessed on 2 February 2023).

### 4.5. RNA-Seq Analysis

P6 and JM198 were tolerant to salt stress and were selected for RNA-Seq analysis. The seedlings of P6 and JM198 were treated using the same evaluation method tested in this study. The leaves of each sample were collected one day after the salt stress treatment. Total RNA was extracted using a RNeasy Plant Mini Kit (Qiagen, Hilden, Germany). The RNA-Seq libraries were constructed using a TruSeq-stranded mRNA library prep kit (Illumina, CA, USA), following the manufacturer’s instructions. The low-quality reads and adaptor sequence were removed from the raw reads using Trimmomatic. The cleaned reads were mapped to the reference genome (IRGSP-1.0) using HISAT2, and the expression level of each gene was obtained by HTSeq-count. The expression level of each gene was presented as the fragments per kilobase per million (FPKM) values. To correct the variation in expression levels between samples, it was normalized using DESeq and used for DEG analysis. The DEGs were selected using DESeq normalization results, according to the following criteria: log_2_ fold change ≥ 1.0, *p*-value < 0.05.

## 5. Conclusions

In this study, we detected four QTLs related to salt tolerance at the seedling stage using GWAS analysis. Of the four QTLs, *qDTS1-2* was the most significant QTL for salt tolerance at the seedling stage. *qDTS1-2*, with the largest −log10(P) value of 5.81, was located between flanking SNPs (1354576 and id1028360) on chromosome 1. Using RNA-Seq analysis, 7 DEGs among the 665, showing different expression levels in tested salt tolerance lines, were eventually identified around the target regions of the 4 QTLs detected in this study. Based on these results, we suggest that the two genes *Os01g0963600* (ASR transcription factor) and *Os01g0975300* (*OsMYB48*) are the most promising candidate genes associated with salt tolerance at the seedling stage.

## Figures and Tables

**Figure 1 plants-12-01401-f001:**
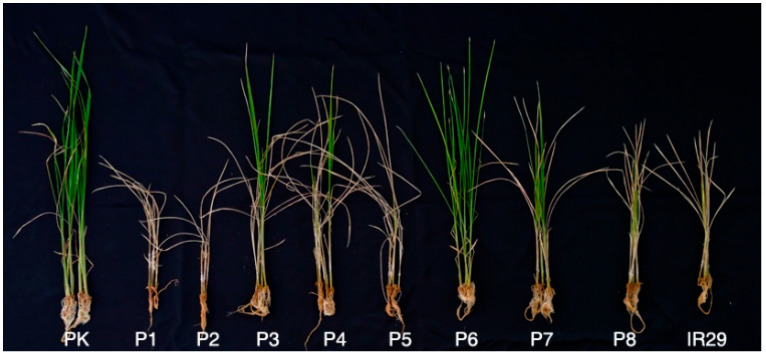
Salt tolerance of the 8 parents by 14 days under 0.7% NaCl stress at the seedling stage. PK: Pokkali (Tolerance check), IR29 (Susceptible check), P1–P8: eight parents.

**Figure 2 plants-12-01401-f002:**
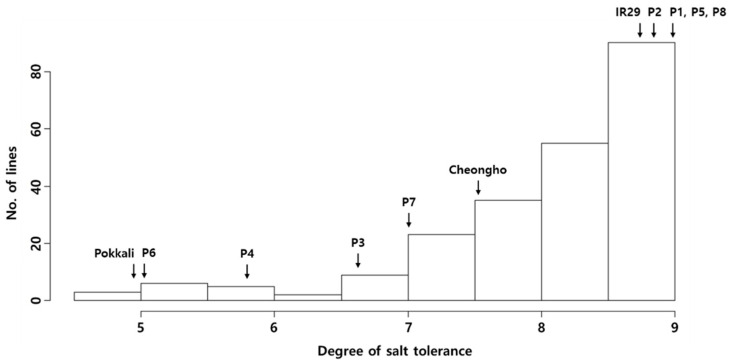
Frequency distribution for the degree of salt tolerance at the seedling stage in the *japonica*-MAGIC population.

**Figure 3 plants-12-01401-f003:**
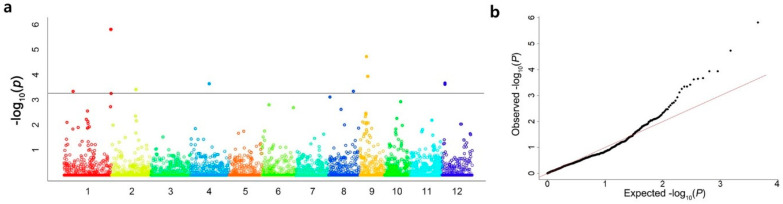
GWAS for salt tolerance at the seedling stage in the *japonica*-MAGIC population. Manhattan plot (**a**) and quantile-quantile plot (**b**) for the GWAS in the *japonica*-MAGIC population using the MLM. The horizontal line indicates the threshold value (*p* = 5.0 × 10^−4^).

**Figure 4 plants-12-01401-f004:**
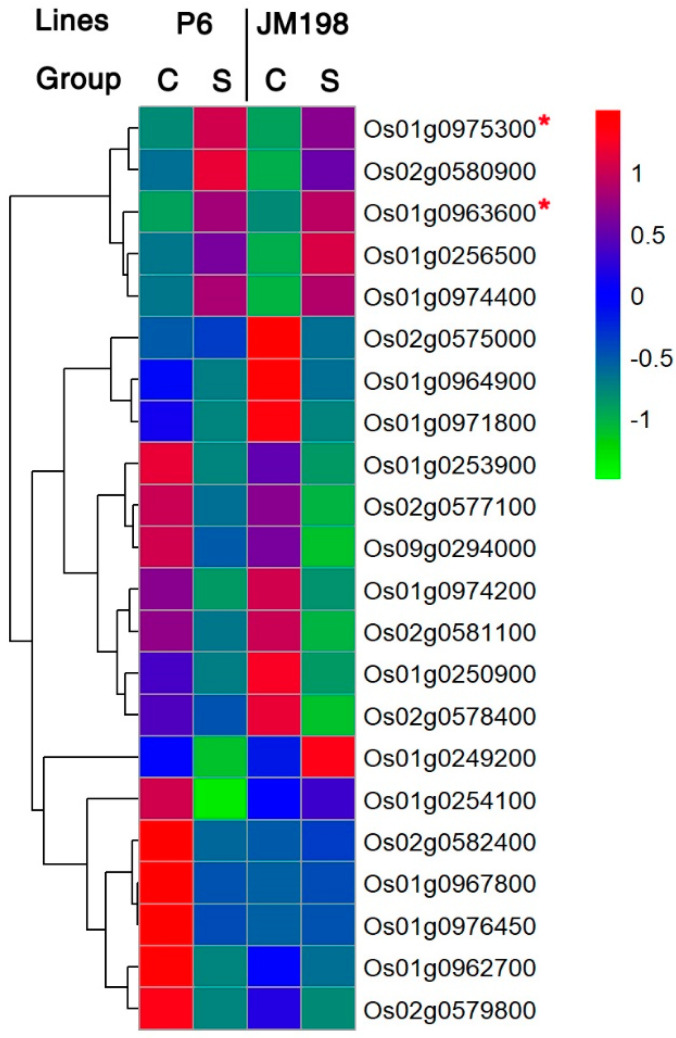
Gene expression analysis of 22 candidate genes related to salt tolerance between P6 and JM198. C: control group, S: salt stress group, *: Commonly upregulated DEGs in both P6 and JM198 located in the target regions of *qDTS1-2*.

**Figure 5 plants-12-01401-f005:**
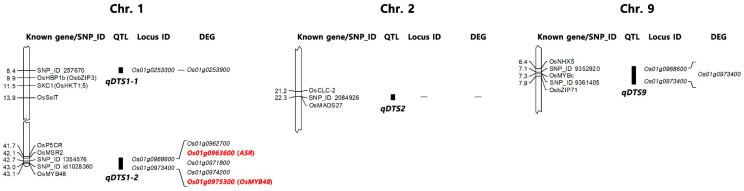
The physical location of QTLs is associated with salt tolerance at the seedling stage on chromosomes 1, 2, and 9, respectively. Known gene/SNP_ID: Known genes related to salt tolerance closely located to the QTLs are represented in the gene symbols, and the SNP marker used for GWAS in this study is represented in SNP_ID. QTL: QTLs related to salt tolerance were identified in this study. Locus ID: Significant SNPs related to salt tolerance were identified by GWAS in this study. DEG: DEGs identified in this study are represented in locus ID. Two upregulated genes under salt stress are represented in red. The physical map distance is represented in Mb.

**Table 1 plants-12-01401-t001:** Evaluation criteria for salt tolerance at the seedling stage in this study.

Degree	Observation	Tolerance
1	Normal growth and no leaf symptoms	Highly tolerant
3	Nearly normal growth, but the leaf tips of a few leaves are whitish and rolled	Tolerant
5	Growth is severely retarded, most leaves rolled, and only a few are elongated	Moderately tolerant
7	Complete cessation of growth, most leaves dry, and some plants dying	Susceptible
9	Almost all plants dead or dying	Highly susceptible

**Table 2 plants-12-01401-t002:** Significant SNPs for salt tolerance at the seedling stage from GWAS analysis.

QTL	SNP_ID	Chr. ^a^	Position	*p*-Values	MAF ^b^	PVE (%) ^c^	FDR ^d^	Locus ID
*qDTS1-1*	257670	1	8385903	4.55 × 10^−4^	0.05	7.65	0.30	*Os01g0253300*
*qDTS1-2*	1354576	1	42720390	1.53 × 10^−6^	0.05	15.20	0.01	*Os01g0968600*
	id1028360	1	42977253	5.56 × 10^−4^	0.11	7.40	0.34	*Os01g0973400*
*qDTS2*	2084926	2	22256108	3.83 × 10^−4^	0.08	7.87	0.30	-
*qDTS9*	9352920	9	7146425	1.15 × 10^−4^	0.05	9.41	0.19	-
	9361405	9	7332926	1.15 × 10^−4^	0.05	9.41	0.19	*Os09g0299200*

^a^ Chr.: chromosome number, ^b^ MAF: minor allele frequency, ^c^ PVE: phenotypic variance explained, ^d^ FDR: false discovery rate.

**Table 3 plants-12-01401-t003:** Differentially expressed candidate genes in P6 and JM198 under salt stress conditions.

QTLs	Chr. ^a^	Locus ID	Gene Function	Log_2_ Fold Change	
				P6 (Control vs. Salt)	JM198 (Control vs. Salt)
*qDTS1-1*	1	*Os01g0253900*	Triacylglycerol lipase	−1.63	−1.55
*qDTS1-2*	1	*Os01g0962700*	Peroxidase 12 precursor	−2.84	−1.41
	1	*Os01g0963600*	ASR transcription factor	2.46	1.91
	1	*Os01g0971800*	Transcription factor with a GARP DNA-binding domain	−5.29	−6.90
	1	*Os01g0974200*	Metallothionein	−2.26	−2.51
	1	*Os01g0975300*	MYB-related transcription factor	2.59	2.72
*qDTS2*	2	-	-	-	-
*qDTS9*	9	*Os09g0294000*	Bifunctional aspartokinase/homoserine dehydrogenase 2	−1.89	−4.04

^a^ Chr.: chromosome number, locus ID, and gene function were searched using RAP-DB (https://rapdb.dna.affrc.go.jp, accessed on 2 February 2023).

## Data Availability

Not applicable.

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
