# Peer review of "Identification of Candidate Genes for Salt Tolerance at the Seedling Stage Using Integrated Genome-Wide Association Study and Transcriptome Analysis in Rice"

_plants, 2023, doi:10.3390/plants12061401_

Round 1

Reviewer 1 Report

The MS “ Identification of candidate genes for salt tolerance at the seedling stage using integrated genome-wide association study and transcriptome analysis in rice” identified two candidate genes related to salt tolerance in rice by using integrated GWAS and transcriptome analysis on MAGIC population. The research design is reasonable, the result is valuable to rice breeding. However, several minor revisions are needed:

1.      The description should be more succinct, such as: L281, “P6 (IR73571-3B-11-3-K2), one of the parents of the JMAGIC population”- P6 was introduced already in previous sentence, no need to repeat again. There are similar situations in other parts also and I will not put it one by one here……

      Moreover, “showed a tolerance reaction” isn’t appropriate due to it didn’t provide clear information, the line has good tolerance or bad tolerance to stress?

2.      Totally, how many lines in magic population?  In L278, it was 381, while in L128, it was 221.

3.      Why selected P6 and JM198 as materials for transcriptome analysis? What’s the parents for line JM198. How about it’s response to salt stress and what’s it DTS?

4.      L287, three replications and L295, seven replications. Please descript clearly.

5.      Figure 1. “P1–P8: 8 parents” can be moved to back of IR29, 8 changed to eight.

Figure 4. caption, “C: control group, T: salt stress group” , the corresponding letters are lost in the figure.

6.      L183-L185, it’s confusing, the 22 genes were reportedly related to salt stress tolerance or others? If it is, please present in a table. Meanwhile, I suggest to do a venn analysis which can present a clear information. Furtherly, the total transcriptome result should be provided in supplementary files. The present supplementary files isn’t enough.

7.      It’s confusing in L213-L217, what’s the reason for different salt response of P1 between present result and previous report?  And how do authors get the conclusion” Therefore, we could confirm P6 as a salt tolerance source conferring tolerance to salt stress in the JMAGIC population.”?

8.      Please revise the English carefully can make it more concise and logical.

Author Response

Dear reviewer,

We thank the reviewer for their useful and constructive comments. We revised our manuscript by responding to all of the comments, which led to substantially improved manuscript. We marked the revised parts in red, and included the additional supplementary tables.

We hope that our revision would be acceptable for the publication in “plants”.

Sincerely Yours,

Suk-Man Kim, Ph.D.

Department of Ecological & Environmental System, Kyungpook National University, Sangju 37224, Republic of Korea

Tel.: +82(54)530-1206

[email protected]

Reviewer 2 Report

Dear, this is a well-written article of scientific interest. My consideration is to accept it for publication.

Author Response

Dear reviewer,

We would like to express our gratitude to the reviewer for their positive evaluation of this paper, which presents candidate genes related to salt tolerance in rice.

We hope that our revision would be acceptable for the publication in “plants”.

Sincerely Yours,

Suk-Man Kim, Ph.D.

Department of Ecological & Environmental System, Kyungpook National University, Sangju 37224, Republic of Korea

Tel.: +82(54)530-1206

[email protected]

Reviewer 3 Report

In the manuscript entitled "Identification of candidate genes for salt tolerance at the seedling stage using integrated genome-wide association study and transcriptome analysis in rice." the authors investigated different rice lines to identify salt tolerance related genes. The research work is very well-planed and performed, the chosen system is adequate for this research. The methods were described in detail, so the experiments can be repeated by others and the presentation of the results are clear. In general, the manuscript is written in good manner and order. The authors made a clear conclusion, they selected 2 promising candidate genes related to salt tolerance.

Author Response

(The authors gave the same response as above.)
